# Dying “from” or “with” COVID-19 during the Pandemic: Medico-Legal Issues According to a Population Perspective

**DOI:** 10.3390/ijerph18168851

**Published:** 2021-08-22

**Authors:** Fabio De-Giorgio, Vincenzo M. Grassi, Eva Bergamin, Alessandro Cina, Franca Del Nonno, Daniele Colombo, Roberta Nardacci, Laura Falasca, Celeste Conte, Ernesto d’Aloja, Gianfranco Damiani, Giuseppe Vetrugno

**Affiliations:** 1Department of Health Care Surveillance and Bioethics, Section of Legal Medicine, Università Cattolica del Sacro Cuore, 00168 Rome, Italy; vincenzograssi@live.com (V.M.G.); eb95@live.it (E.B.); celestemgconte@gmail.com (C.C.); giuseppe.vetrugno@policlinicogemelli.it (G.V.); 2Fondazione Policlinico Universitario A. Gemelli, IRCCS, 00168 Rome, Italy; alessandro.cina@policlinicogemelli.it (A.C.); Gianfranco.Damiani@unicatt.it (G.D.); 3Department of Diagnostic Imaging, Oncological Radiotherapy and Hematology, Diagnostic Imaging Area, Università Cattolica del Sacro Cuore, 00168 Rome, Italy; 4Pathology Unit, National Institute for Infectious Diseases “Lazzaro Spallanzani”, IRCCS, 00149 Rome, Italy; franca.delnonno@inmi.it (F.D.N.); daniele.colombo@inmi.it (D.C.); 5Laboratory of Electron Microscopy, National Institute for Infectious Diseases “Lazzaro Spallanzani”, IRCCS, 00149 Rome, Italy; roberta.nardacci@inmi.it (R.N.); laura.falasca@inmi.it (L.F.); 6Department of Medical Sciences and Public Health, University of Cagliari, 09124 Cagliari, Italy; ernestodaloja@gmail.com; 7Department of Life Sciences and Public Health, Università Cattolica del Sacro Cuore, 00168 Rome, Italy

**Keywords:** COVID-19, post-mortem computed tomography, CO intoxication, pneumonia, causality

## Abstract

There is still a lack of knowledge concerning the pathophysiology of death among COVID-19-deceased patients, and the question of whether a patient has died *with* or *due* to COVID-19 is still very much debated. In Italy, all deaths of patients who tested positive for SARS-CoV-2 are defined as COVID-19-related, without considering pre-existing diseases that may either contribute to or even cause death. Our study included nine subjects from two different nursing homes (Cases 1–4, Group A; Cases 5–9, Group B). The latter included patients who presumably died from CO poisoning due to a heating system malfunction. All subjects tested positive for COVID-19 both ante- and post-mortem and were examined using post-mortem computed tomography prior to autopsy. COVID-19 was determined to be a contributing cause in the deaths of four out of nine subjects (death *due* to COVID-19; i.e., pneumonia and sudden cardiac death). In the other five cases, for which CO poisoning was identified as the cause of death, the infection presumably had no role in exitus (death *with* COVID-19). In our attempt to classify our patients as dying with or due to COVID-19, we found the use of complete assessments (both histological analyses and computed tomography examination) fundamental.

## 1. Introduction

In December 2019, an outbreak of lower respiratory tract infection cases was detected in Wuhan City in the Hubei Province of China. In the absence of a known etiological agent, the first cases were classified as “pneumonia of unknown etiology.” Following careful investigations by the local CDC (Center for Disease Control and Prevention), the etiology of the disease was attributed to severe acute respiratory syndrome coronavirus 2 (SARS-CoV-2). The novel coronavirus disease was named “coronavirus disease 19” (COVID-19) and declared a pandemic by the World Health Organization (WHO) in March 2020. To date, over 127.3 million cases and over 2.7 million deaths have been confirmed worldwide [1]. COVID-19 infection represents a complex challenge and continuous threat to public health. Although many studies regarding this topic can be found in the literature, there is still a general lack of knowledge concerning the pathophysiology of death among deceased patients. Undoubtedly, COVID-19-related signs and symptoms are extremely variable, and the clinical course of the disease can vary from person to person; nonetheless, studies concerning the causes and underlying mechanisms of death are insufficient, and this automatically translates into an unrealistic assessment of infection-related mortality rates. As a consequence, problems may arise both from an epidemiological point of view and from a clinical point of view, with doctors having to face unpleasant situations; indeed, a lack of information regarding the potential causes of death can hinder the implementation of correct prevention and treatment measures in specific patients. In this regard, autopsies can provide clinically relevant insights and support, and should be mandatory in order to identify not only the cause of death, but also the underlying infection-related mechanisms and patterns of organ damage (i.e., with the use of biopsies). Needless to say, this can only be achieved by following specific guidelines and recommendations on the best standards of practice and safety measures [2]. Studies have demonstrated that autopsy personnel have a minimal risk of acquiring infection, provided specific protocols are followed and precautions are taken. Thus, in the case of COVID-19-positive bodies, autopsies should be considered safe and, in fact, encouraged, because they have been shown to confirm the presence of COVID-19 even after extensive periods of time post-mortem [3,4].

As mentioned, the clinical course of COVID-19 is highly heterogeneous; the majority of patients are either asymptomatic or experience only mild symptoms (i.e., fever, dry cough, shortness of breath) [2]. In contrast, in some cases, the disease causes severe respiratory symptoms, often requiring oxygenation and admission to an intensive care unit (ICU), with symptoms and signs of multi-organ manifestations. This extensive involvement can be partially explained by the high tropism demonstrated by the virus towards angiotensin-converting enzyme II (ACE2). ACE2 receptors are ubiquitously expressed in the body (lungs, vessels, liver, kidneys, spleen, skin, muscle, and nervous structures), and their binding to SARS-CoV-2 represents an essential step for the viral invasion of human cells [5]. In a study by Zou et al., the authors identified the organs and specific cell types (with ACE2 expression) at risk for SARS-CoV-2 infection: lung (type II alveolar cells), heart (myocardial cells), kidney (proximal tubule cells), ileum and esophagus (epithelial cells), bladder (urothelial cells), venous and arterial endothelial cells, and arterial smooth muscle cells [5,6]. Such multi-organ involvement can be studied and evaluated by analyzing the course of the disease from a clinical and radiological point of view, and by integrating these findings with the valid support of complete post-mortem examinations and histological analysis, which provide valuable information through the analysis of transformative phenomena and organ damage.

In the literature, there are several reports concerning the increased risk of cardiovascular (CV) complications connected to COVID-19. The two conditions appear to be closely related; acute respiratory infections (including COVID-19) are known to act as triggering factors for CV diseases (CVD) [7,8] and, vice versa, pre-existing CVD and other comorbidities usually increase the incidence of infectious diseases [9]. Indeed, SARS-CoV-2 infection has been associated with cases of acute myocardial infarctions (MIs), myocarditis, arrhythmias, and venous thromboembolism [10].

Herein, we report nine cases of SARS-CoV-2 infection-positive patients in two different groups in order to discuss these cases in terms of “*deaths with COVID-19 or deaths due to COVID-19*”.

## 2. Materials and Methods

Between March 2020 and January 2021, two groups of corpses arrived at our institute from two different nursing homes where COVID-19 outbreaks were suspected. Our study was carried out on a total of 9 cases, including patients from 2 different nursing homes. Based on the provenance of the bodies, we divided the subjects into two groups: Cases from 1 to 4 were assigned to Group A (these corpses which arrived at our institute from nursing home A) and Cases from 5 to 9 were assigned to Group B (these corpses which arrived at our institute from nursing home B) (Table 1). The conducted investigations, including total-body CT examination and complete autopsy (macroscopic and microscopic examination, toxicological analyses), were authorized by the Judicial Authority.

The study was approved by the Institutional Research Ethics Committee (ID 3862).

The mean age in Group A was 83 years (range: 72–91 years). The mean age in Group B was 87 years (range: 71–99 years). All patients in both groups were female. In Group B, all 5 patients presumably died at night from carbon monoxide poisoning due to a heating system malfunction; the cadavers were found in the same nursing home. All 9 subjects had undergone routine COVID-19 testing (in order to exclude a cluster in Group A, and as a screening organized by the National Health Service in Group B) and, in all cases, positivity to the virus was confirmed post-mortem by means of RT-PCR on swabs performed at the level of the right and left bronchi and right and left pulmonary parenchyma. All cases were examined, prior to autopsy, with the use of post-mortem total-body CT. Afterwards, all corpses underwent a complete autopsy, including the collection of specimens for histological evaluation based on standard H&E staining.

### 2.1. Toxicological Analysis

Toxicological analyses were carried out for all Group B cases. In 2/5 cases, peripheral blood and urine samples were analyzed. In the remaining 3 cases, samples were collected from the peripheral blood and vitreous humor, because urine was either contaminated or not available.

The presence of ethanol was investigated by adding sodium chloride and an internal standard n-propanol to 0.2 mL of peripheral blood. The sample was then analyzed using headspace gas chromatography. A reference sample at a concentration of 1 g/L was also examined (parameters: Finnigan trace GC gas chromatograph with FID detector, Supelcowax10 column 20 m, id 0.25 mm phase 0.25 um. Temperature: 60 °C (1’), 35 °C/min at 220 °C, 5 min). Moreover, in order to evaluate the presence of toxic substances or drugs, urine and blood samples were analyzed using liquid chromatography and high-resolution mass spectrometry.

The percentage of carboxyhemoglobin was calculated via spectrophotometry on a whole blood sample using the “Huffner–Heilmeyer” coefficient method. In total, 20 μL of blood was hemolyzed with 5 mL of 1% ammonia solution and directly subjected to spectrophotometric analysis (“Evolution 201” Thermo Double Beam Spectrophotometer).

### 2.2. Post-Mortem CT

Each corpse was examined with the use of PMCT prior to autopsy. The scans were conducted on a Somatom Scope 16-slice CT scanner, Siemens Healthineers Italia, and characterized by the following parameters: 130 kV, 150 mA, 2.4 mm slice thickness. The corpses were placed on a horizontal CT table in a supine position with their arms at their sides. They were fully clothed and wrapped in body bags. Whole-body CT scans were obtained. No contrast agent was used in this procedure.

## 3. Results

The results have been summarized in Table 2. All corpses from Group A displayed findings of interstitial pneumonia, ischemic heart disease (chronic ischemic heart disease, N = 3; recent ischemic necrosis and myocardial fibrosis, N = 1), and some degree of cardiac hypertrophy. Case 1, in which signs of recent ischemic necrosis were observed, also showed evidence of micro-thrombosis and myocarditis upon histopathological examination. In this case, the cause of death was identified as SCD, sustained by the rupture of an atherosclerotic plaque in a subject with signs of multiorgan involvement (i.e., lung). Cases 2 to 4, on the other hand, died of COVID-19 pneumonia and displayed suggestive pulmonary findings, including pulmonary edema, fibrous thickening of the septa, and basal lung hepatization (Figure 1). Concerning Group B cases, even though all patients tested positive for COVID-19 both ante- and post-mortem, the cause of death was CO intoxication. All corpses showed similar autopsy and histological findings, including characteristic cherry-red hypostasis and internal visceral cherry-red coloration, pulmonary edema and congestion with intra-alveolar hemorrhages, and various levels of atherosclerosis (Figure 2).

Typical CT imaging of COVID-19, including bilateral multi-lobar ground glass opacities (GGO), were seen in all cases (Figure 3). A peripheral localization of GGO was found only in Group A, Case 3. In Group A, no gravity-dependent distribution of GGO was identified in Cases 1, 3 or 4; in Group B, it was only found in Case 6. Alveolar consolidations were found in two cases for both groups. Lymphadenopathies were identified in only one case. Pleural effusions were commonly found in both groups (three of four in Group A and five of five in Group B). CT findings are reported in Table 3.

There was a suspicion of CO intoxication; therefore, toxicological analyses were performed in all the Group B cases, with the aim of determining both the cause and manner of death. The search for ethanol via headspace gas chromatography yielded negative results in all five cases. Concerning toxic substances/drugs, we found promazine in the therapeutic range (N = 1), benzodiazepines and metabolites in the therapeutic range (i.e., flurazepam and alprazolam) (N = 1), and quetiapine and metabolite at subtherapeutic levels (N = 1). In terms of carboxyhemoglobin (CO-Hb) percentages, these were found to be 40% (N = 2), 45–50% (N = 1), and 50–55% (N = 2). These values appear to be compatible with severe carbon monoxide poisoning. According to the literature, values higher than 40–50% are almost invariably associated with a clinical picture of severe poisoning, including lethargy, coma, motor incoordination, memory, and praxia disorders [11], as well as the possible instability of cardiac, respiratory, and neurological functions, up to death [12].

## 4. Discussion

The mortality rate of SARS-CoV-2 infection varies from country to country, probably as a result of various factors such as different public health services and adopted policies [13]. In Italy, the Italian National Institute of Health (ISS, Istituto Superiore di Sanità) has established a surveillance system with the aim of gathering data concerning all COVID-19 cases throughout the country [14,15]. COVID-19-associated deaths were defined as deaths in patients who had tested positive for the virus (fatality rate: number of deaths in persons who tested positive for SARS-CoV-2 divided by the number of SARS-CoV-2 cases) [14].

Initially, the case–fatality rate in Italy was higher than that observed in other countries [15], and this could be explained by different factors. First of all, it is important to consider the fact that Italy has the highest percentage of elderly population in Europe, and the second-highest percentage worldwide after Japan; in this regard, it has been proven that a strong correlation exists between the severity and risk of death of SARS-CoV-2 infection and age [16]. In addition, COVID-19 mortality was found to be dependent on the presence of serious comorbidities, which are rather prevalent in the Italian population [17]. Finally, the high case–fatality rate may also be explained by the methods used to identify COVID-19–related deaths by the Italian NHS. Indeed, in Italy, all deaths of patients who tested positive for SARS-CoV-2 are defined as COVID-19-related, without taking into account pre-existing diseases that may either contribute or even cause death [15]. Hence, this system may have led to an overestimation of the overall case–fatality rate. 

In the context of legal medicine, the question of whether a patient has died *with* or *due to* COVID-19 is very much debated, especially considering the implications that this particular matter has in terms of professional responsibility. Coroners are frequently asked to assess whether COVID-19 played a direct role in a patient’s death, as well as whether the medical staff’s treatment and management were appropriate. Indeed, medical malpractice cases are not uncommon. Early on in the pandemic, we witnessed an increase in the risk of infection, as well as the overall number of positive cases; one of the reasons for this was that many COVID-19-positive patients were relocated to nursing homes and assisted living facilities due to a shortage of hospital beds [18,19,20]. Furthermore, data regarding the disease, its treatment, and preventive measures were limited. As a result, dealing with professional liability issues has represented a complex task for coroners; if we then consider the question of “dying with or due to COVID-19”, the matter certainly becomes even more problematic. In this setting, despite the high risk of infection for personnel, autopsies play a critical role, because they determine the severity of direct viral damage and assess whether organ failure plays a part in a patient’s death. Multiple data sources, such as ante- and post-mortem microbiological testing, CT scans, complete forensic autopsy, and histological analyses, may be relevant in this regard [20].

Differences in terms of surveillance methods also play a fundamental role when it comes to answering the query. It may be helpful to consider a “*death due to COVID-19*” as one resulting from a compatible illness, in either a suspected or confirmed SARS-CoV-2-positive case, in the absence of a clear alternative cause, such as trauma [21]. Thus, it is safe to indicate as the underlying cause of death when the latter takes part in a causal chain that leads to death [22]. In a study by Cobos-Siles et al., the authors evaluated 128 individuals with the aim of differentiating between patients whose death was directly associated with the development of COVID-19 complications and SARS-CoV-2-positive patients who died of causes unrelated to COVID-19 complications [23]. Among the complications associated with COVID-19, the authors included acute respiratory distress syndrome (ARDS) or hyperinflammation response, acute respiratory failure, severe lung injury on chest X-ray/CT, thromboembolic events, and septic shock. On the other hand, “*deaths with COVID-19*” were defined as those deaths in which the disease acted as a precipitating factor, leading to the decompensation of underlying pathologies. According to the authors, only 20% of patients died of unrelated complications [24]. Similarly, Slater et al. examined 162 COVID-19-positive cases and found the disease to be the direct cause of death in 150 (93%) patients [25]. Grippo et al. analyzed a total of 5311 death certificates of persons who had tested positive for SARS-CoV-2 infection [26]. According to the authors, COVID-19 was the underlying cause of death in 88% of cases.

Regarding our study population, Case 1 died of sudden cardiac death (SCD) sustained by the rupture of an atherosclerotic plaque (middle third of the anterior descending coronary). The subject tested positive for COVID-19 and displayed signs of multiorgan involvement (lung, heart, liver). The underlying mechanisms that could explain Case 1’s clinical picture are as follows: (a) T cell- and monocyte-mediated hyperinflammation and cytokine storm that leads to myocarditis; (b) respiratory failure and hypoxia, especially in the setting of ARDS and severe infections, resulting in the damage of cardiac muscle cells; (c) down-regulation of angiotensin-converting enzyme 2 (ACE2) expression and subsequent protective signaling pathways in cardiac myocytes; (d) blood hypercoagulability and the development of coronary microvascular thromboses; (e) diffuse endothelial lesions and inflammation of the endothelium in various organs (including the heart) as either a direct consequence of SARS-CoV-2 viral involvement and/or resulting from the host’s inflammatory response; and (f) inflammation and/or stress causing coronary plaque rupture or supply–demand mismatch, leading to myocardial ischemia/infarction [27,28].

In a study by Sheth et al., the authors hypothesized that the cytokine storm may play a role in the pathogenesis of acute coronary syndromes. In particular, the authors identified two possible pathophysiological mechanisms: on the one hand, systemic inflammation caused by the viral infection can determine a rapid formation of a coronary plaque or an acute modification of an existing plaque, causing the rupture of its surface with exposure of the underlying components that determine the formation of a thrombus; on the other hand, systemic viral infection can cause direct myocardial damage, both through coronary vasoconstriction and the stimulation of platelet activation, as well as an increased prothrombotic condition, either mediated by the cytokine storm or caused by a hypoxia-related increase in metabolic demand [29]. Based on such considerations, it was possible to assume that SARS-CoV-2 infection and the consequent onset of COVID-19 may have played a con-causal role in the occurrence of the rupture of the atherosclerotic plaque and, ultimately, in the death of Case 1.

In Case 2, there was evidence of a more severe pulmonary involvement. Thus, the lung was the organ that was mainly affected by the infection, which inevitably played a causal role in exitus. Likewise, the deaths in Cases 3 and 4 were attributed to COVID-19-related interstitial pneumonia in subjects suffering from chronic ischemic heart disease. As mentioned, most COVID-19-positive patients display mild symptoms or are completely asymptomatic. Nonetheless, approximately 20% of affected patients deteriorate rapidly and develop severe respiratory diseases, such as pneumonia, with significant mortality rates [30]. Studies have determined that a correlation exists between the development of major complications/outcomes and both increasing age and a high number of associated comorbidities [30,31]. Advanced age, as for many other diseases and infections, is a known risk factor for COVID-19 and its complications; it has been observed that the majority of patients affected by severe COVID-19 pneumonia are elderly and that, of all the predictors of mortality in COVID-19 pneumonia, advanced age is the most important [32,33]. This correlation can be explained by the physiological deterioration of one’s immune system that occurs with age, as well as by the age-dependent loss of T cells and T cell subtypes [34,35]. Lymphopenia, increased CRP, and erythrocyte sedimentation were observed in the laboratory results of COVID-19 patients [36]. The word “lymphopenia” refers to a reduction in the peripheral lymphocyte count (primarily CD4 + T and CD8 + T cells); this condition increases the risk of secondary bacterial infections [37].

Conversely, Cases 5 to 9 refer to deaths from CO poisoning. Indeed, all corpses displayed pulmonary findings indicative of both the cause and mechanism of death. Carboxyhemoglobin in the blood is responsible for the development of characteristic autopsy findings, such as “cherry-red” hypostases (that typically appear at carboxyhemoglobin concentrations >30%), marked pulmonary edema, parenchymal imbibition, and severe polyvisceral congestion. CO causes harm to the lungs via direct hypoxic damage to tissues and pathological alterations that may be secondary to CO’s effect on the heart [38]. Thus, cases of CO intoxication often display intense pulmonary hyperemia and congestion, acute emphysema, interstitial thickening, focal infiltrations of polymorphonuclear leukocytes, focal and diffuse endo-alveolar hemorrhages, and epithelial desquamation [39]. These findings were observed in all our cases (5 to 9) and was further supported by histopathological examination. The results of our analysis are in accordance with those observed by King et al. [40]. Traces of pigment were appreciable in the majority of subjects. Three out of five showed desquamation of the bronchial epithelium, with the presence of necrotic material; bronchial necrosis (usually mucosal necrosis) may indicate exposure to toxic substances and/or fumes [41]. Considering COVID-19, lung damage plays a central role; findings typically include interstitial pneumonia, ground glass pattern (honeycombing pattern), marked edema and congestion, diffuse alveolar damage (DAD) and endo-alveolar hemorrhages, extensive micro-thrombotic phenomena and vascular wall damage, formation of hyaline membranes, and, in some cases, ARDS [42,43,44]. Furthermore, damage often occurs ubiquitously within the organism, with variable expression of the disease. In Cases 5 to 9, autopsy findings were consistent with what has been described in the literature about COVID-19-related organ damage. However, although such findings are compatible with COVID-19, they are virtually always present in conditions that cause lung injury and respiratory distress, and this makes them highly nonspecific. Thus, we concluded that, in the five aforementioned subjects, COVID-19 had no role in the causal chain that led to their deaths. We can describe them as five relatively healthy subjects with age-related pathologies who presented no infection-related complications and in whom both the cause and mechanism of death were attributable to CO poisoning. According to us, Cases 5 to 9 fall within the definition of “deaths *with* COVID-19 infection”.

As a general rule, biological samples (i.e., nasopharyngeal swabs) performed at autopsy help avoid forensic consequences from misdiagnosis and allow us to identify those alterations that are only visible at the microscopic level. Thus, pathologists may provide only temporary diagnoses based solely on macroscopic evidence; the latter should be subsequently confirmed by histological analysis [45,46,47,48,49]. This is particularly relevant if we consider our case-series and the post-mortem CT findings; in fact, there was a heterogeneous appearance, without clear signs helpful to distinguish between parenchymal alteration related to COVID-19 infection and CO poisoning. In forensic medicine, one is always wary of the use of conclusions concerning the assessment of the existence of a causal link, based solely on the chronological criterion, or, worse, on the criterion of proximate cause, as illustrated by the Latin locution “post hoc *ergo propter hoc*”. Recently, many of the discussions regarding whether the deaths recorded in the COVID-19 era were deaths *from* or *with* COVID-19 still seem to be contaminated by the processes of intuitive reasoning, dictated by the “post hoc *ergo propter hoc*” fallacy. Nonetheless, forensic medicine has always promoted the cultivation of doubt and the use of fruitful, counterintuitive reasoning.

The clinical definition of *ascertained* COVID-19 cases is based on the following criterion: a positive molecular swab in a patient with a clinical picture suggestive of COVID-19 (in particular, the respiratory function is affected, as well as other systems and mechanisms, such as coagulation, with the latter being triggered by respiratory distress or consequent hospitalization). If this criterion applies to the in vivo assessment of COVID-19, chances are that it may also apply to its post-mortem diagnosis. If death should occur months after the clinical assessment of COVID-19, any connection with COVID-19 can only be assumed, and therefore be considered valid, if it meets the “phenomenological continuity” criterion (i.e., if an ascertained continuum of temporal and clinical–laboratory results exists, without which the diagnosis of death *from* COVID-19 cannot in any way be made). For instance, consider the case of a COVID-19-positive subject who dies from a multidrug-resistant superinfection following a prolonged hospitalization for the management of COVID-19-related clinical problems. The swab performed on their corpse tests negative to the infection. In this case, the superinfection will certainly be ascribed as a contributing cause to the death of the subject; however, the primary cause will be traced back to COVID-19. Considering a subject with a respiratory, clinical, and radiological picture suggestive of COVID-19, a negative serology for common germs but constantly negative swabs, their death will not be defined as *confirmed death from COVID-19* but as *probable death from COVID-19*; this reasoning must necessarily be applied to any investigation carried out on a corpse displaying these same characteristics.

The dead speak for themselves, but their language, in particular with regard to the taxonomy of COVID-19, cannot be radically different from that of the living, considering both confirmed and probable cases of COVID-19.

## 5. Conclusions

The difficulties encountered when trying to differentiate between deaths *with* COVID-19 and deaths *due to* COVID-19 represent a substantial problem, especially considering the fact that the concept of causality is fundamental in legal medicine. For epidemiological purposes, it would be advisable to perform nasopharyngeal swabs and PMCT scans in all suspected cases in order to distinguish those in which SARS-CoV-2 infection caused the death from those where COVID-19 was only part of the causal link. Pre-autopsy nasopharyngeal swabs can help us understand whether a corpse is infected or not; likewise, PMCT can be useful for diagnostic purposes (identification of characteristic COVID-19 CT findings) and, especially in the early hours post-mortem, it can provide information regarding the extent and severity of the disease [50,51].

Moreover, the need for a differential diagnosis between deaths *with* and deaths *due to* COVID-19 has several implications in terms of penal liability. Forensic pathologists, who are often asked to express themselves in terms of the cause of death, are required to establish a causal link between the infection and the death of a person. In this regard, although it may be time-consuming and costly, the need for complete assessments (both autopsy and ancillary exams) is evident; one must neither spare expenses nor consider the benefit/risk ratio when the need to satisfy both justice and public health is at stake.

## Figures and Tables

**Figure 1 ijerph-18-08851-f001:**
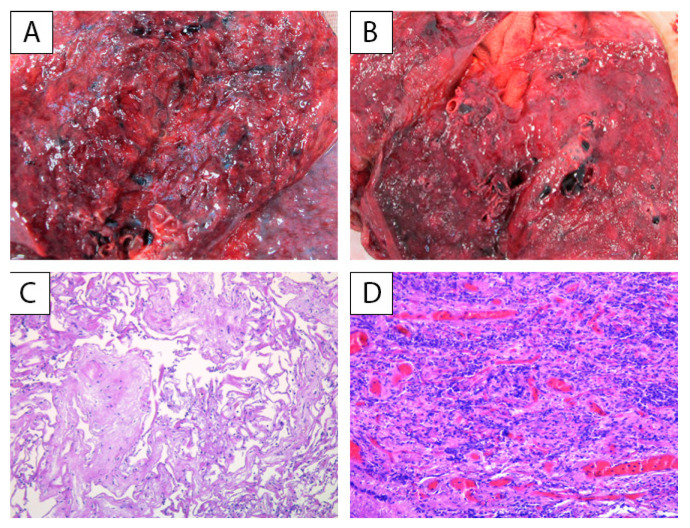
Group A: lung macroscopic and microscopic findings: (**A**,**B**) cut surface showing the consolidation of lobes and red congested areas, with thickening of the interstitial septa and pulmonary edema; (**C**) interstitial fibrosis H&E, 200×; (**D**) numerous inflammatory cells consisting mostly of lymphocytes infiltrating into alveolar septa and clustering around capillary vessels, H&E, 200×.

**Figure 2 ijerph-18-08851-f002:**
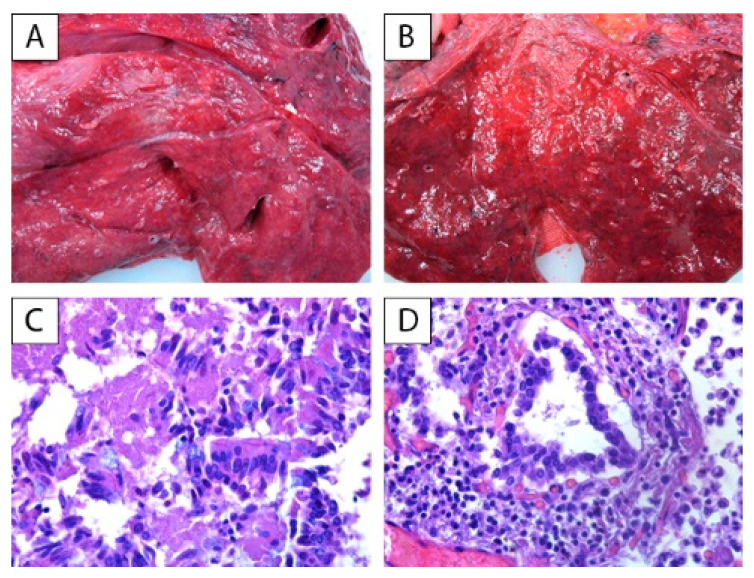
Group B: pulmonary macroscopic and microscopic findings: (**A**,**B**) cherry-red coloration, pulmonary edema, and congestion; compact basal parenchyma; (**C**) desquamative necrosis of the bronchial epithelium, H&E, 400×; (**D**) bronchi surrounded by inflammatory cells, H&E, 400×.

**Figure 3 ijerph-18-08851-f003:**
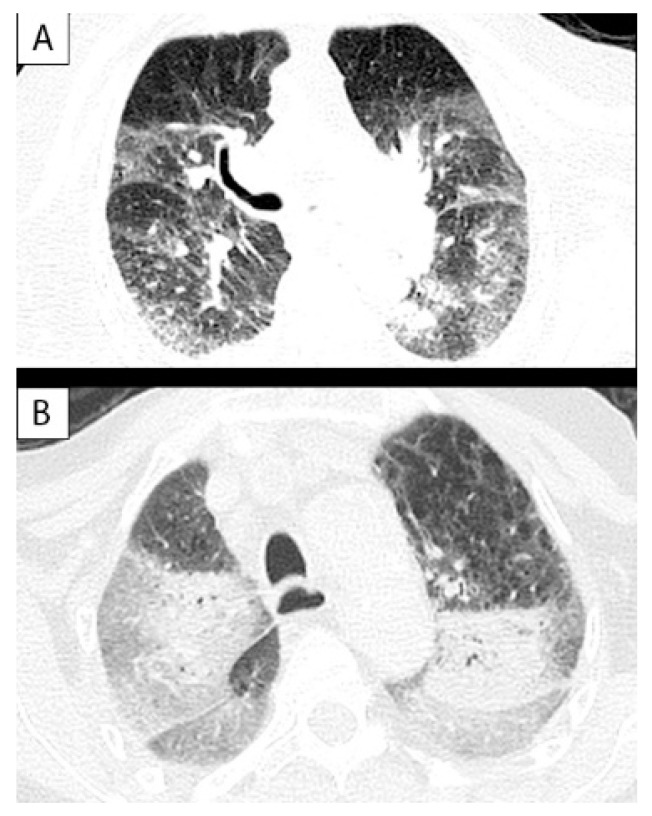
Post-mortem lung CT (parenchymal window) showing “ground glass” opacities in a subject of group A (**A**) and in one of group B (**B**). In A, the distribution of opacities was bilateral, peripheral, and without gravitational gradient. This pattern is considered typical in COVID-19 pulmonary infection. In B, the “ground glass” opacities showed a more central and gravitational distribution, as seen in post-mortem CT lung changes.

**Table 1 ijerph-18-08851-t001:** Clinical characteristics of the study population.

Groups	Case	Age	Gender	Comorbidity
Group A	1	72	F	Diabetes; Chronic cerebral vasculopathy.
	2	79	F	Right parietal ischemic stroke; Epilepsy; Secondary parkinsonism; Senile dementia.
	3	90	F	Not available
	4	91	F	Not available
Group B	5	89	F	Diabetes; Hypo-mobility.
	6	87	F	Dilated cardiomyopathy (ICD); Hypertension; Atrial fibrillation; COPD; Obesity; Chronic renal failure.
	7	99	F	COPD; Hypertension; Chronic renal failure; Diabetes; Polyarthritis.
	8	89	F	Hypertension; Polyarthritis; Cerebrovascular leukoencephalopathy.
	9	71	F	Not available

**Table 2 ijerph-18-08851-t002:** Post-mortem findings and causes of death.

Groups	Case ^#^	Autopsy	Histology	Cause of Death
Group A	1	Eccentric cardiac hypertrophy; LAD chronic atherosclerosis with thrombosis; Apical emphysema, compact basal parenchyma, edema.	Interstitial pneumonia; Micro-thrombosis; Myocarditis; Recent ischemic necrosis in myocardial fibrosis; Dilation of the hepatic centrilobular veins.	Sudden cardiac death in COVID-19
	2	Concentric cardiac hypertrophy; moderate atherosclerosis; Pulmonary edema, basal hepatization, fibrous thickening of the septa.	Interstitial pneumonia; Chronic ischemic heart disease.	COVID-19 pneumonia
	3	Concentric cardiac hypertrophy; moderate atherosclerosis Pulmonary edema, basal hepatization, fibrous thickening of the septa;	Interstitial pneumonia and honeycomb appearance; Chronic ischemic heart disease; Dilation of the hepatic centrilobular veins.	COVID-19 pneumonia
	4	Septal hypertrophy; slight atherosclerosis; Pulmonary oedema, fibrous thickening of the septa.	Interstitial pneumonia; Chronic ischemic heart disease.	COVID-19 pneumonia
Group B	5	Cherry-red hypostasis and internal visceral cherry-red coloration; Concentric cardiac hypertrophy; slight atherosclerosis; Pulmonary edema and congestion.	Chronic ischemic heart disease; Giant cell bronchiolitis, endo-alveolar hemorrhages.	CO intoxication (COHb 55%)
	6	Cherry-red hypostasis and internal visceral cherry-red coloration; Slight atherosclerosis; Pulmonary edema and congestion.	Chronic ischemic heart disease; COPD; endo-alveolar hemorrhages.	CO intoxication (COHb 50%)
	7	Cherry-red hypostasis and internal visceral cherry-red coloration; Septal hypertrophy; moderate atherosclerosis; aortic ectasia; Pulmonary oedema, congestion, and compact basal parenchyma.	Chronic ischemic heart disease; Interstitial lung fibrosis; endo-alveolar hemorrhages.	CO intoxication (COHb 55%)
	8	Cherry-red hypostasis and internal visceral cherry-red coloration; Concentric cardiac hypertrophy; moderate atherosclerosis; aortic ectasia; Pulmonary edema and congestion.	Chronic ischemic heart disease; Pulmonary edema; intra-alveolar hemorrhages; bronchial necrosis.	CO intoxication (COHb 40%)
	9	Cherry-red hypostasis and internal visceral cherry-red coloration; Moderate atherosclerosis; myocardial scars; Pulmonary edema and congestion	Chronic ischemic heart disease; Pulmonary fibrosis and edema, intra-alveolar hemorrhages and pigment-laden macrophages.	CO intoxication (COHb 40%)

**Table 3 ijerph-18-08851-t003:** Post-mortem CT findings.

CT Finding	Case 1	Case 2	Case 3	Case 4	Case 5	Case 6	Case 7	Case 8	Case 9
Ground glass opacities	x	x	x	x	x	x	x	x	x
-Peripheral			x						
-Bilateral	x	x	x	x	x	x	x	x	x
-Multilobar	x	x	x	x	x	x	x	x	x
-no gravitational distribution	x		x	x		x			
“Crazy paving” pattern			x						
Alveolar consolidations		x	x		x				x
Lymphadenopathies			x						
Pleural effusion	x	x	x		x	x	x	x	x

## Data Availability

The data presented in this study are available on request from the corresponding author.

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
