# Peer review of "Dying “from” or “with” COVID-19 during the Pandemic: Medico-Legal Issues According to a Population Perspective"

_ijerph, 2021, doi:10.3390/ijerph18168851_

Round 1
Reviewer 1 Report
The manuscript is interesting and gives an important contribution to the forensic field community to answer a research dilemma between the deaths from Covid and deaths with Covid. Some revisions need to be made:
1) The abstract should also be improved by inserting materials and methods and results;
2) In the materials and methods, it should be specified:
- a) The division criterion between Group A and Group B.
- b) Post-mortem CT should be included in materials and methods as a separate paragraph, also specifying the type of device.
- c) The modality of the toxicological analysis in cases of death from carbon monoxide poisoning should be inserted briefly. Was this type of analysis done in all or only in some of the cadavers? Is there any circumstantial data that prompted forensic pathologists to carry out a toxicological investigation?
3) Figures 1, 2, and 3 should be moved to the results paragraph.
4) Panel C figure 1 should be eliminated because the Masson's trichrome was not in the materials and methods, and the reason for using this histological stain is not specified.
5) In the introduction it should be specified how the autopsy is a safe procedure for providing information on COVID-19, citing this new manuscript: Safe Management Strategies in Clinical Forensic Autopsies of Confirmed COVID-19 Cases. Diagnostics 2021, 11, 457. https://doi.org/10.3390/diagnostics11030457,
and
Post-Mortem Detection of SARS-CoV-2 RNA in Long-Buried Lung Samples. Diagnostics (Basel). 2021 Jun 24;11(7):1158. doi: 10.3390/diagnostics11071158. PMID: 34202678.
Author Response
We welcome the Reviewers’ comments and suggestions and have modified the manuscript according to their advice (the changes are highlighted in red). Please find our itemized, point-by-point response to the comments of the Reviewers below, with the indication of the changes we made.
The Manuscript has also has undergone English language editing by MDPI (see attached certificate).
ANSWERS TO REVIEWER COMMENTS.
The manuscript is interesting and gives an important contribution to the forensic field community to answer a research dilemma between the deaths from Covid and deaths with Covid. Some revisions need to be made:
1) The abstract should also be improved by inserting materials and methods and results;
- The abstract has been strengthened according to Reviewer’s suggestions.
2) In the materials and methods, it should be specified:
- a) The division criterion between Group A and Group B.
- b) Post-mortem CT should be included in materials and methods as a separate paragraph, also specifying the type of device.
- c) The modality of the toxicological analysis in cases of death from carbon monoxide poisoning should be inserted briefly. Was this type of analysis done in all or only in some of the cadavers? Is there any circumstantial data that prompted forensic pathologists to carry out a toxicological investigation?
- We provided the requested information within the text.
3) Figures 1, 2, and 3 should be moved to the results paragraph.
- The figures have been placed in the results paragraph.
4) Panel C figure 1 should be eliminated because the Masson's trichrome was not in the materials and methods, and the reason for using this histological stain is not specified.
- Figure 1C has been replaced; instead of a Masson's trichrome, a microphotograph showing interstitial fibrosis in H&E staining has been presented.
5) In the introduction it should be specified how the autopsy is a safe procedure for providing information on COVID-19, citing this new manuscript: Safe Management Strategies in Clinical Forensic Autopsies of Confirmed COVID-19 Cases. Diagnostics 2021, 11, 457. https://doi.org/10.3390/diagnostics11030457,
and
Post-Mortem Detection of SARS-CoV-2 RNA in Long-Buried Lung Samples. Diagnostics (Basel). 2021 Jun 24;11(7):1158. doi: 10.3390/diagnostics11071158. PMID: 34202678.
- The introduction has been modified according to the Reviewer’s suggestions and the references have been inserted within the text.

Reviewer 2 Report
This is a well-written case report on deaths as a direct result of Covid-19 or as a side effect of Covid-19. It has implications in medico-legal issues but this was not discussed very-well in the discussion or introduction section. Author's used nine human cases, but it's not clear how they got access to these cases, and whether any approval for human subjects was obtained. Method section needs more detailed information on how data were analyzed. It is not clear what RT-PCR kit was used for the confirmation of Covid-19 positive/negative cases, and what instrument was used for total-body CT. It is also not clear how histological slides were prepared. Result section need a summary of major findings from Table 2, and the discussion section needs to focus more on results from this study.
Author Response
We welcome the Reviewers’ comments and suggestions and have modified the manuscript according to their advice (the changes are highlighted in red). Please find our itemized, point-by-point response to the comments of the Reviewers below, with the indication of the changes we made.
The Manuscript has also has undergone English language editing by MDPI (see attached certificate).
This is a well-written case report on deaths as a direct result of Covid-19 or as a side effect of Covid-19. It has implications in medico-legal issues but this was not discussed very-well in the discussion or introduction section. Author's used nine human cases, but it's not clear how they got access to these cases, and whether any approval for human subjects was obtained.
- The required information have been provided. We specified that the corpses arrived at our Institute from two distinct nursing homes where a COVID-19 outbreak was suspected. In addition, we mentioned that all conducted investigations, including the total-body CT examination and complete autopsy (macroscopic and microscopic examination, toxicological analyses) were authorized by the Judicial Authority and that the study was approved by the Institutional Research Ethics Committee (ID 3862).
Method section needs more detailed information on how data were analyzed. It is not clear what RT-PCR kit was used for the confirmation of Covid-19 positive/negative cases, and what instrument was used for total-body CT. It is also not clear how histological slides were prepared.
- The Materials and Methods section has been implemented.
Result section need a summary of major findings from Table 2, and the discussion section needs to focus more on results from this study.
- The results section now contains a brief description of the results. The discussion section has been modified, examining more in detail the results.

Reviewer 3 Report
Dear authors,
Main proposal of the article is absolutely interesting. Despite it, i think the study design can only partially (and poorly) answer your question and fit the aim.
Great concerns belong to the common cause of death of Group B. In other words, it looks to be too easy to obtain the outcome "death with covid" in these cases. I can't found great differences between these CO intoxications and another cause of death with a quick onset - traumatic - not related to pathologies (for ex. if the 5 people would have died for a firarms wound on the head being covid-positive). In this last case (of the firarms wound in positive people) you, of course, would not have great issues to consider them as deaths with covid, and so you should in your article.
Now that being said, I found the proposal of answering and discuss the question in the title is not possibile with your approach. It could be possible if you had a group of people dead with a covid positivity, but for pathologies to which you were able to reconduct the death uniquely (by autopsy-histological and/or CT findings).
So, if you don't have a similiar sample, I have to suggest a complete study re-design: it may be, at most, a case report in which you demonstrate how you made your diffencial diagnosis between covid related pattern of pneumonia (macro-histo and CT) and pattern of CO intoxication. Otherwise, consider a low soundness, as testimonial and circumnstantial evidence, together with the toxicological determination of CO concentration, would allow you to make this differential diagnosis in a such easy way.
Title should change and fit the study design. Text should be signifcantly shorten. Most of the introduction and the first part of discussion (the one before starting talking about your cases) is unecessary and contains well know notions about COVID, so it should be shorten. Case 1 shouldn't be considered.
In table 3, "ct findings" should be on the upper row.
Author Response
We welcome the Reviewers’ comments and suggestions and have modified the manuscript according to their advice (the changes are highlighted in red). Please find our itemized, point-by-point response to the comments of the Reviewers below, with the indication of the changes we made.
The Manuscript has also has undergone English language editing by MDPI (see attached certificate).
Dear authors,
Main proposal of the article is absolutely interesting. Despite it, i think the study design can only partially (and poorly) answer your question and fit the aim.
Great concerns belong to the common cause of death of Group B. In other words, it looks to be too easy to obtain the outcome "death with covid" in these cases. I can't found great differences between these CO intoxications and another cause of death with a quick onset - traumatic - not related to pathologies (for ex. if the 5 people would have died for a firarms wound on the head being covid-positive). In this last case (of the firarms wound in positive people) you, of course, would not have great issues to consider them as deaths with covid, and so you should in your article.
Now that being said, I found the proposal of answering and discuss the question in the title is not possibile with your approach. It could be possible if you had a group of people dead with a covid positivity, but for pathologies to which you were able to reconduct the death uniquely (by autopsy-histological and/or CT findings). So, if you don't have a similiar sample, I have to suggest a complete study re-design: it may be, at most, a case report in which you demonstrate how you made your diffencial diagnosis between covid related pattern of pneumonia (macro-histo and CT) and pattern of CO intoxication. Otherwise, consider a low soundness, as testimonial and circumnstantial evidence, together with the toxicological determination of CO concentration, would allow you to make this differential diagnosis in a such easy way.
Title should change and fit the study design. Text should be signifcantly shorten. Most of the introduction and the first part of discussion (the one before starting talking about your cases) is unecessary and contains well know notions about COVID, so it should be shorten. Case 1 shouldn't be considered.
In table 3, "ct findings" should be on the upper row.
- Dear Reviewer, the idea for this paper originated from the arrival at our Institute of two distinct groups of deceased patients, both from nursing homes where a COVID-19 outbreak was suspected, just a few months apart. The trait d’union of these two groups was therefore the suspect of a cluster of “deaths due to COVID-19”, with relevant penal consequences, not only related to involuntary manslaughter, but also non-intentional spread of a dangerous epidemic disease. Despite the anamnestic mention of hypothesis of CO exposition, the Prosecutor asked for a full autopsy to ascertain the cause of death. We compared the two groups by using PMCT, macroscopic, and microbiological data first; we then waited a few days for the toxicologist to provide the results. Obviously, this is not a case-control study; it should be considered as a case series starting from a common suspicion based on an epidemiological analysis performed in two nursing homes.
- We tried to strengthen the whole manuscript keeping in mind your suggestion in the hope that this new version could be suitable for publication
- Case 1 has not been excluded because it could represent a valid example of con-cause, especially in the Italian Penal Law interpretation. Table 3 has been modified as suggested.

Reviewer 4 Report
Taking into account the introduction and most part of discussion, it would seem that the study wants to help shed light on the problem of cause of death in Covid-19 positive subjects: dying "from" or "with" Covid-19 during the pandemic, as the headline states. This objective would have required the presentation of a large and polymorphic case study and not just 9 cases, of which 5 died from carbon monoxide poisoning. The design of the study is therefore inadequate with respect to the objectives it wants to achieve. On the other side, important in the work is the proposal to submit the bodies of the deceased to the computed tomography examination before the autopsy. I therefore recommend resetting the work by increasing the case series if the authors want to discuss on the cause of death (from or with) in Covid-19 deceased patients.
Author Response
We welcome the Reviewers’ comments and suggestions and have modified the manuscript according to their advice (the changes are highlighted in red). Please find our itemized, point-by-point response to the comments of the Reviewers below, with the indication of the changes we made.
The Manuscript has also has undergone English language editing by MDPI (see attached certificate).
Taking into account the introduction and most part of discussion, it would seem that the study wants to help shed light on the problem of cause of death in Covid-19 positive subjects: dying "from" or "with" Covid-19 during the pandemic, as the headline states. This objective would have required the presentation of a large and polymorphic case study and not just 9 cases, of which 5 died from carbon monoxide poisoning. The design of the study is therefore inadequate with respect to the objectives it wants to achieve. On the other side, important in the work is the proposal to submit the bodies of the deceased to the computed tomography examination before the autopsy. I therefore recommend resetting the work by increasing the case series if the authors want to discuss on the cause of death (from or with) in Covid-19 deceased patients.
- Dear Reviewer, the idea for this paper originated from the arrival at our Institute of two distinct groups of deceased patients, both from nursing homes where a COVID-19 outbreak was suspected, just a few months apart. The trait d’union of these two groups was therefore the suspect of a cluster of “deaths due to COVID-19”, with relevant penal consequences, not only related to involuntary manslaughter, but also non-intentional spread of a dangerous epidemic disease. Despite the anamnestic mention of hypothesis of CO exposition, the Prosecutor asked for a full autopsy to ascertain the cause of death. We compared the two groups by using PMCT, macroscopic, and microbiological data first; we then waited a few days for the toxicologist to provide the results. Obviously, this is not a case-control study; it should be considered as a case series starting from a common suspicion based on an epidemiological analysis performed in two nursing homes.We are aware of the potential pitfalls and limitations of case reports and case series; nonetheless, their importance has been emphasized in the Literature.
- We tried to strengthen the whole manuscript keeping in mind your suggestion in the hope that this new version could be suitable for publication
De-Giorgio F, Grassi VM, Miscusi M, Mancuso C, d'Aloja E, Pascali VL. Subarachnoid hemorrhage and carbon monoxide exposure: accidental association or fatal link? J Forensic Sci. 2013 Sep;58(5):1364-6. doi: 10.1111/1556-4029.12168. Epub 2013 May 17. PMID: 23683314.
Rossi R, Lodise M, Lancia M, Bacci M, De-Giorgio F, Cascini F. Trigemino-cardiac reflex as lethal mechanism in a suicidal fire death case. J Forensic Sci. 2014 May;59(3):833-5. doi: 10.1111/1556-4029.12408. Epub 2014 Feb 6. PMID: 24502511.
De Giorgio F, Arena V. Case reports: importance and problems. Forensic Sci Int. 2007 May 24;168(2-3):e54; author reply e55. doi: 10.1016/j.forsciint.2007.02.012. Epub 2007 Mar 23. PMID: 17383131.
Madea B. Case histories in forensic medicine. Forensic Sci Int. 2007 Jan 17;165(2-3):111-4. doi: 10.1016/j.forsciint.2006.05.012. Epub 2006 Jun 21. PMID: 16793230.

Round 2
Reviewer 4 Report
The reported problems of this paper have not yet been resolved. The study design is not adequate for the goals the authors want to achieve. The revised form added unnecessary parts. The title of this study should be changed and the paper reviewed and proposed as a case report, taking into account that group A can not be compared with group B because they are very different. Basically Group A and B share the same pre-existing cardiac diseases, but for group A death from Covid-19 can be identified in interstitial pneumonia which was not present in any subject of Group B, whose death is clearly due to CO poisoning. Therefore, any deduction on death “from” or “with” Covid-19 starting from the comparison of the two groups is inconsistent. One of the most important data that emerges from this investigation is the failure of postmortem CT in the diagnosis, given that bilateral and multilobar ground-glass opacity, believed to be pathognomonic for Covid-19 lung disease, were present in both groups, with and without interstitial pneumonia. This evidence should deserve more consideration by the authors.
The authors should rewrite the work as case-report. To this end, I suggest a drastic reduction of the paper:
- Introduction: cut from “Studies have demonstrated …”to “… venous thromboembolism (8)”;
- Materials and Methods: reduce/cut toxicological analysis;
- Results: drastically reduce the results of toxicological analysis
- Discussion: cut the first part from “The mortality rate …” to “…fatality rate”;
- Discussion: cut from “Differences in terms …” to “… 88% of cases.”;
- Discussion: cut from “The underlying mechanisms…” to “…ischemia/infarction (19-20)”;
- Discussion: cut from “As mentioned … “ to “…bacterial infections (41).” (text added in red by the authors).
- Discussion: cut from “Carboxy-hemoglobin … “ to “… histopathological examination” (text added in red by the authors).
- Discussion: cut from “Considering COVID-19 …” to “… highly nonspecific.” (text added in red by the authors).
Author Response
.